# Evaluation of the ZWD/ZTD Values Derived from MERRA-2 Global Reanalysis Products Using GNSS Observations and Radiosonde Data

**DOI:** 10.3390/s20226440

**Published:** 2020-11-11

**Authors:** Liangke Huang, Lijie Guo, Lilong Liu, Hua Chen, Jun Chen, Shaofeng Xie

**Affiliations:** 1College of Geomatics and Geoinformation, Guilin University of Technology, Guilin 541004, China; guolj@glut.edu.cn (L.G.); lllong99@glut.edu.cn (L.L.); xieshaofeng@glut.edu.cn (S.X.); 2Guangxi Key Laboratory of Spatial Information and Geomatics, Guilin University of Technology, Guilin 541004, China; 3School of Geodesy and Geomatics, Wuhan University, Wuhan 430079, China; ttchen@whu.edu.cn (H.C.); cj_snake@whu.edu.cn (J.C.)

**Keywords:** zenith tropospheric delay, zenith wet delay, GNSS, MERRA-2

## Abstract

Tropospheric delay is one of the main errors affecting high-precision positioning and navigation and is a key parameter of water vapor detection in the Global Navigation Satellite System (GNSS). The second Modern-Era Retrospective analysis for Research and Applications (MERRA-2) is the latest generation of reanalysis data collected by the National Aeronautics and Space Administration (NASA), which can be used to calculate tropospheric delay products with high spatial and temporal resolution. However, there is no report analyzing the accuracy of the zenith tropospheric delay (ZTD) and zenith wet delay (ZWD) calculated from MERRA-2 data. This paper evaluates the performance of the ZTD and ZWD values derived from global MERRA-2 data using global radiosonde data and International GNSS Service (IGS) precise ZTD products. The results are as follows: (1) Taking the precision ZTD products of 316 IGS stations from around the world from 2015 to 2017 as the reference, the average root mean square (RMS) of the ZTD values calculated from the MERRA-2 data is better than 1.35 cm, and the accuracy difference between different years is small. The bias and RMS of the ZTD values show certain seasonal variations, with a higher accuracy in winter and a lower accuracy in summer, and the RMS decreases from the equator to the poles. However, those of the ZTD values do not show obvious variations according to elevation. (2) Relative to the radiosonde data, the RMS of the ZWD and ZTD values calculated from the MERRA-2 data are better than 1.37 cm and 1.45 cm, respectively. Furthermore, the bias and RMS of the ZWD and ZTD values also show some temporal and spatial characteristics, which are similar to the test results of the IGS stations. It is suggested that MERRA-2 data can be used for global tropospheric vertical profile model construction because of their high accuracy and good stability in the global calculation of the ZWD and ZTD.

## 1. Introduction

Tropospheric delay is a key factor affecting high-precision spatial positioning, such as the Global Navigation Satellite System (GNSS), very long baseline interferometry (VLBI) and satellite laser ranging (SLR), which are also the basic data in atmospheric scientific exploration [1,2,3,4]. In GNSS data processing, the empirical tropospheric delay model is generally used to calculate zenith hydrostatic delay (ZHD) or zenith tropospheric delay (ZTD) information, which is taken as a prior value, and the zenith wet delay (ZWD) or ZTD residual error and position parameters are taken as unknown parameters to be solved [5]. Relevant studies have shown that the accuracy of the empirical tropospheric delay model also has a certain impact on the positioning results and positioning speed of GNSS, especially in GNSS precise point positioning (GNSS-PPP), which can greatly affect the estimation accuracy and convergence time of elevation components [5,6,7,8]. However, with the continuous improvement in the GNSS positioning algorithm, the ZTD information from regional or global GNSS stations with high precision and high time resolution can be obtained through GNSS precision data processing. The International GNSS Service (IGS) and Crustal Movement Observation Network of China (CMONOC) provide global users with precise ZTD information for free, which in turn can provide important data support for high-precision GNSS applications. Although many GNSS reference stations have been built around the world, these stations are not evenly distributed, they are all located in land areas, and there is a lack of GNSS reference stations in ocean areas. In addition, although the observation data of GNSS reference stations have a very high time resolution, it is inevitable that there will be missing data, and much of the data from some of the reference stations are missing. Therefore, it is obvious that the precise ZTD information provided by the global GNSS reference station is not the optimal choice in the selection of data sources for the construction of a global high-precision and high-resolution tropospheric delay model. However, atmospheric reanalysis data, such as National Centers for Environmental Prediction (NCEP) reanalysis data and European Center for Medium-Range Weather Forecasts (ECMWF) reanalysis data, can be used to calculate ZWD and ZTD information and have received widespread attention [9,10].

Thus far, a variety of atmospheric reanalysis data have been provided internationally, such as ECMWF reanalysis data and NCEP reanalysis data. Related scholars have used ECMWF or NCEP atmospheric reanalysis data to construct regional or global tropospheric delay models [11,12,13,14,15,16,17]. Moreover, many Chinese scholars have constructed a wealth of regional or global tropospheric delay correction models using atmospheric reanalysis data [18,19,20,21,22,23,24,25]. In addition, the use of atmospheric reanalysis data to improve the positioning accuracy and convergence efficiency of GNSS-PPP has also become an important topic [26,27].

Moreover, related research has also been carried out in the use of atmospheric reanalysis data to calculate the accuracy evaluation of tropospheric delay information. Chen et al. [28] evaluated the accuracy of the ZTD values calculated from ECMWF/NCEP reanalysis data with different spatial resolutions in China using Global Positioning System (GPS) observations from CMONOC. The results show that the accuracy of the ZTD calculated from ECMWF reanalysis data is significantly better than that calculated from NCEP reanalysis data. In addition, Chen et al. [29] evaluated the accuracy of the ZTD derived from ECMWF/NCEP reanalysis data of Asia using the ZTD values estimated by 49 GPS stations and obtained meaningful conclusions. The results illuminated the accuracy and feasibility of computing the tropospheric delays and establishing the ZTD prediction model over Asia for navigation and positioning with ECMWF and NCEP data. Wang et al. [30] evaluated the accuracies of precipitable water vapor (PWV) values from five reanalysis products. The evaluation results showed that the reanalysis products with the best PWV accuracy is the fifth generation ECMWF Reanalysis (ERA5). Jiang et al. [31] evaluated the latest ERA5 atmospheric reanalysis data released by the ECMWF using ZTD data from 219 GNSS stations in the CMONOC. The results showed that the ERA5 data have extremely high accuracy in calculating the ZTD by integration in China, and its accuracy is better than that of ERA-Interim data. In general, the accuracy of atmospheric reanalysis data needs to be evaluated by independent observations before application. The second Modern-Era Retrospective analysis for Research and Applications (MERRA-2) is the latest atmospheric reanalysis data provided by National Aeronautics and Space Administration (NASA), which has a very high spatial and temporal resolution [32,33], but until now, no report has evaluated the accuracy of the ZWD and ZTD calculated from the MERRA-2 atmospheric reanalysis data by the integration method on a global scale. Therefore, the accuracy of the ZWD and ZTD calculated from MERRA-2 data by the integration method is verified by combining global radiosonde data and IGS precise ZTD products. These data are important for the subsequent study of the construction of a global vertical profile function model using MERRA-2 data, so it has important practical significance.

This paper is organized as follows. The data and the methodology for ZTD and ZWD calculation from MERRA-2 are introduced in Section 2. The accuracies of the ZWD and ZTD are shown, and the global temporal and spatial variation features of the ZWD and ZTD biases and root mean square (RMS) are analyzed in Section 3. The summary and conclusions of this paper are given in Section 4.

## 2. Data and Methodology

### 2.1. Data

#### 2.1.1. Reanalysis Products

MERRA-2 data are the latest atmospheric reanalysis data provided by the National Aeronautics and Space Administration (https://goldsmr4.Gesdisc.eosdis.nasa.gov/data/MERRA2). The horizontal resolution is 0.5° × 0.625°, and the vertical resolution has 42 pressure levels reaching 0.1 hPa at the top level. The temporal resolution of the pressure-level data is 6 h, and the temporal resolution of the surface data is 1 h. This paper evaluates only the ZTD and ZWD calculated by integration from the pressure-level MERRA-2 data. Therefore, pressure data (i.e., the air pressure, temperature, specific humidity and geopotential height and the corresponding surface meteorological parameters and surface elevation) with a resolution of 6 h must be used for each grid point in the world.

#### 2.1.2. Radiosonde Data

The University of Wyoming freely provides global radiosonde observations (http://weather.uwyo.edu/upperair/sounding.html). Currently, there are more than 1500 radiosonde stations globally. The radiosonde profiles, with a resolution of 12 h, i.e., at UTC 00:00 and 12:00 every day, mainly include pressure-level profiles and surface observations. In this study, 409 radiosonde stations are selected worldwide (Figure 1). The radiosonde-derived pressure and ZWD are used to calculate the ZTD and are used as reference values to evaluate the ZTD calculated from the MERRA-2 data. Moreover, the ZWD values derived from the radiosonde data are also used to assess the ZWD values calculated from the MERRA-2 data.

#### 2.1.3. GNSS Observations

The International GNSS Service archives and distributes zenith total delay estimates from the GNSS observation data at all the IGS stations with a temporal resolution of 5 min and with a latency of approximately 4 weeks [34] (ftp://cddis.gsfc.nasa.gov/pub/gps/products/). The global ZTD products collected by the IGS are accurate to 4 mm and can be used as a reference standard to evaluate other ZTD products [35]. This paper selects a ZTD time series of more than three hundred IGS stations in the world, and the locations of each station are shown in Figure 1.

### 2.2. Methods

#### Deriving the ZTD and ZWD from MERRA-2 Meteorological Data at GNSS Stations and Radiosonde Stations

The derivation of the ZTD and ZWD from pressure-level data requires two steps: calculating the zenith tropospheric delay and zenith wet delay by the integration method from Level 1 to Level 42 and calculating the tropospheric delay above the top level using the Saastamoinen model to obtain comparable results [28]. Then, the ZTD and ZWD of the GNSS stations and radiosonde stations were obtained by interpolation on the basis of the ZTD and ZWD of the grid points.

Generally, the grid-point ZTD and ZWD values are calculated based on the MERRA-2 pressure-level data using the integral method with the following formula [36,37]:(1)$$N=k_{1}\;\times \left(P-e\right)/T+k_{2}\;\times e/T+k_{3}\times \;e/T^{2}$$
(2)$$N_{w}=k_{2}\times \;e/T+k_{3}\times \;e/T^{2}$$
(3)e=h×P/0.622
where $k_{1}$ = 77.604 K/Pa, $k_{2}\;$ = 64.79 K/Pa, $k_{3}$ = 377,600.0 K^2^/Pa, *P* represents the atmospheric pressure, *h* the specific humidity, *N* the total refraction, $N_{w}$ the wet refraction, and *e* the vapor pressure. Then, the ZTD and ZWD values are calculated by the following formulas [38]:(4)$$\mathrm{ZTD}=10^{-6}\overset{\;}{\underset{s}{\int }}Nds=10^{-6}\underset{i}{\sum }N_{i}∆s_{i}$$
(5)$$\mathrm{ZWD}=10^{-6}\overset{\;}{\underset{s}{\int }}N_{w}ds=10^{-6}\underset{i}{\sum }N_{w}{}_{i}∆s_{i}$$

The Saastamoinen model is used to calculate the zenith delay above the top level, and the meteorological data of the top level are used as the input values of the model [39],
(6)$$\mathrm{ZTD}=0.0022793\times \frac{\left[P+\left(0.05+\frac{1255}{T}\right)\times e\right]}{f\left(\varphi ,H\right)}$$
(7)$$f\left(\varphi ,H\right)=1-0.00266\cos\left(2\varphi \right)-2.8\times 10^{-7}H$$
where *T* represents the temperature at the top, *P* represents the atmospheric pressure, *e* represents the vapor pressure, φ represents the latitude, and *H* represents the ellipsoidal height of the site.

The locations of the grid points from the MERRA-2 data are different from the locations of GNSS sites and radiosonde sites, and the grid points of the MERRA-2 atmospheric data grid are not consistent with the elevation datum of the GNSS station or radiosonde station. Therefore, it is necessary to obtain the meteorological parameters required in Formulas (1)–(6) by interpolation, and the elevation reference system must be unified before interpolation. The height systems adopted by the MERRA-2, GNSS station and radiosonde station data are the geopotential height, geodetic height and orthometric height, respectively. In the correction of the tropospheric atmospheric elevation, the atmospheric difference caused by the height datum difference between the geopotential height and the orthometric height can be ignored [40]. However, the height system difference between the geopotential height and geodetic height cannot be ignored, which can be calculated by the Earth Gravity Model 2008 (EGM 2008) model to realize the unification of the MERRA-2 data grid points and the GNSS station elevation system [41]. Second, there is a large difference between the grid-point elevation of the MERRA-2 data and that of the GNSS and radiosonde stations, which is very significant in high-altitude areas. Therefore, if the commonly used inverse distance weighting method or bilinear interpolation method is directly used to interpolate the GNSS and radiosonde station data, it will inevitably lead to large interpolation errors, which will affect the accuracy of the evaluation results from MERRA-2 [28]. Therefore, in the integral calculation of ZWD and ZTD from MERRA-2 data, this paper proposes that the altitude of GNSS and radiosonde station should be taken as the starting height for integration to calculate the ZWD and ZTD information of the nearest four grid points of each station. It can be ensured that the height of the four grid points is consistent with the height of the GNSS and radiosonde station, which greatly eliminates or weakens the influence of the ZWD and ZTD on the elevation. However, before performing the integration calculation, vertical interpolation (interpolation or extrapolation) is needed to obtain the meteorological parameters at the height of the GNSS and radiosonde station. In terms of the vertical interpolation, if the elevation of the GNSS and radiosonde station is greater than that of the grid point, the meteorological parameters of adjacent layers can be used for interpolation; otherwise, extrapolation is used. For the vertical interpolation of temperature, the nearest three-layer data at the height of the GNSS and radiosonde station are used to calculate the decline rate in the vertical temperature. If the calculated vertical temperature decline rate is greater than 10 K/km or is a negative value, the temperature decline rate is taken as −6.5 K/km; otherwise, a large interpolation error will be introduced [42,43]. In terms of the vertical interpolation of the atmospheric pressure, the average decline rate of the atmospheric pressure is calculated by using the nearest three-layer data at the height of the GNSS and radiosonde station [44]. In terms of the vertical interpolation of the specific humidity, one grid point (31.5° N, 130.0° E) of MERRA-2 data is selected to analyze the change in specific humidity by height. The results are shown in Figure 2. Figure 2 shows that the variation in the specific humidity by elevation can be approximately expressed by a piecewise linear function. Therefore, the vertical interpolation of the specific humidity can also be calculated by linear interpolation using the data of the nearest three layers at the height of the GNSS and radiosonde station.

According to the above method, the ZWD and ZTD information of the grid point of MERRA-2 data at the height of the GNSS and radiosonde station is calculated, and the horizontal interpolation of the grid point with the inverse distance weighting (IDW) method can achieve a better interpolation effect [45]. Finally, the ZWD and ZTD information at the GNSS and radiosonde station can be obtained by interpolation.

Finally, with the global IGS station precision ZTD product from 2015 to 2017 and the global radiosonde station data from 2015 as reference values, we evaluated the accuracy of the ZWD and ZTD derived from the MERRA-2 data, and the bias and RMS were used as accuracy indicators. The formula are as follows [22]:(8)$$\;\mathrm{bias}=\frac{1}{N}\overset{N}{\underset{i=1}{\sum }}\left(X_{m}^{M_{i}}-X_{m}^{R_{i}}\right)$$
(9)$$\mathrm{RMS}=\sqrt{\frac{1}{N}\overset{N}{\underset{i=1}{\sum }}{\left(X_{m}^{M_{i}}-X_{m}^{R_{i}}\right)}^{2}}$$
where $X_{m}^{M_{i}}$ is the value calculated from the MERRA-2 data, $X_{m}^{R_{i}}$ is the reference value, and *N* represents the number of samples.

## 3. Results and Discussion

### 3.1. Accuracy Comparison of the MERRA-2 ZTD and IGS ZTD

The precise ZTD products with a 5 min time resolution and better than 5 mm precision from 316 IGS stations in the world from 2015 to 2017 were used to test the accuracy of ZTD calculated from MERRA-2 data. The daily average bias and RMS of the integral ZTD of MERRA-2 data at all the IGS stations are calculated, and then the average annual bias and RMS of each IGS station in the world are obtained. The results are shown in Table 1 and Figure 3.

Table 1 shows that the accuracy of the ZTDs calculated from the MERRA-2 data in different years is relatively stable. The bias ranges from −1.49 to 2.47 cm with a mean value of 0.44 cm. The RMS ranges from 0.39 to 2.90 cm with a mean value of 1.32 cm. Figure 3 shows that the bias of the ZTDs calculated from the MERRA-2 data is mainly positive in the middle- and high-latitude regions and negative in some low-latitude regions, such as the east coast of South America and the Indian Ocean. This finding indicates that the MERRA-2 data estimate that the ZTD is larger in middle- and high-latitude regions and smaller in some low-latitude regions. The RMSs of the ZTDs estimated by the MERRA-2 data are smaller in the middle- and high-latitude regions, especially in the Antarctic region. Although it is relatively large in low-latitude regions, its RMS is still approximately 2.0 cm, which may be due to the comprehensive influence of complex climate systems such as tropical rainforest, tropical ocean and tropical steppe climates. The results show that the MERRA-2 data have high accuracy and stability in calculating the ZTD information globally, which is better than the accuracy of the Global Geodetic Observing System (GGOS) atmospheric grid ZTD products, and the global RMS is approximately 1.73 cm [24].

To analyze the daily average variation in the bias and RMS of the ZTD values calculated from the MERRA-2 data, six representative IGS stations, namely, KELY, SYOG, GOL2, DUND, BJCO and MAL2, were selected from the low-, middle- and high-latitude regions of the global Northern and Southern Hemispheres, and the daily average bias and RMS values were statistically analyzed. The results are shown in Figure 4 and Figure 5, and the detailed statistical result are shown in Table 2.

From Figure 4 and Figure 5, Table 2, it can be seen that the daily average bias and RMS of the KELY and GOL2 stations located in the high-latitude and middle-latitude regions of the Northern Hemisphere show obvious seasonal variations, especially for the KELY station, which has a larger bias and RMS during summer and a significant positive bias value. These figures show that the ZTD calculated from the MERRA-2 data in the middle- and high-latitude regions of the Northern Hemisphere is easily affected by the more active water vapor variation in summer. However, there is no obvious seasonal variation in the daily average bias and RMS of the BJCO station in the low-latitude region, but it is relatively large during most of the year, mainly due to the complex climate in the low-latitude region. The DUND and SYOG stations in the middle- and high-latitude regions of the Southern Hemisphere have relatively small daily average biases and RMSs, and there are no obvious seasonal variations. The main reason is that most of the regions in the middle- and high-latitude regions of the Southern Hemisphere are marine regions, and the changes in the meteorological parameters are relatively stable. The daily average bias for the MAL2 station in the low-latitude region shows obvious seasonal variations, and the daily average bias appears larger in winter. In addition, the RMS also shows a relatively large value during most of the year, and there is no obvious seasonal change, which is consistent with the above findings.

To further analyze the monthly average variation in the bias and RMS of the ZTD calculated from the MERRA-2 data, the bias and RMS of the ZTD values calculated for all 316 stations in the world were calculated on a monthly basis. The results are shown in Figure 6.

From Figure 6, it can be seen that the bias of the ZTD values calculated from the MERRA-2 data has obvious seasonal variations, which shows positive bias in all months of the year. The smallest bias of ZTD is 0.23 cm in the winter months and the largest bias of ZTD is 0.72 cm in the summer months. The RMS of the ZTD calculated from the MERRA-2 data is similar to the bias and shows obvious seasonal variations. The smallest RMS of ZTD is 1.07 cm in the winter months and the largest RMS of ZTD is 1.58 cm in summer months, indicating that other meteorological parameters, such as water vapor, are relatively active in the summer months, which has a certain impact on the accuracy of the MERRA-2 data. In other words, the monthly average bias and RMS of the ZTD values calculated from the MERRA-2 data show obvious seasonal variations, but the maximum monthly average RMS is only approximately 1.5 cm, which further indicates that the ZTD values calculated from the MERRA-2 data show good seasonal performance.

Relevant studies have shown that the ZTD is significantly correlated with elevation and latitude. To analyze the variation in the bias and RMS of the ZTD values calculated from the MERRA-2 data, the variation distributions in the bias and RMS in terms of the elevation of the 316 global IGS stations are shown in Figure 7.

Figure 7 shows that the bias and RMS of the ZTD values calculated from the MERRA-2 data have no obvious variation relationship with elevation compared with IGS precise ZTD products. The bias ranges from −1.28 to 2.47 cm with a mean value of 0.43 cm. The RMS ranges from 0.39 to 2.90 cm with a mean value of 1.35 cm. The absolute bias is basically kept within 1.0 cm, and the RMS error is kept within 2.5 cm for all the elevations. This finding supports the paper proposes using the elevation of IGS station as the starting height to calculate the ZTD values of four grid points of MERRA-2 around the IGS station directly, which has achieved good results and greatly weakened the error introduced by ZTD interpolation by elevation.

To further analyze the variation in the bias and RMS of the ZTD values calculated from the MERRA-2 data by latitude, the variation distributions of the bias and RMS of the 316 global IGS stations are shown in Figure 8.

It can be seen from Figure 8 that the bias of the ZTD values calculated from the MERRA-2 data is relatively stable in the global middle- and high-latitude regions, and most of the stations have a positive bias. The bias ranges from −1.28 to 2.47 cm with a mean value of 0.43 cm. The RMS ranges from 0.39 to 2.90 cm with a mean value of 1.35 cm. The bias distribution in the low-latitude regions fluctuates greatly, but in the global latitude distribution, the bias of most of the IGS stations ranges from −1.0 cm to 1.0 cm. The RMS of the ZTD values calculated from the MERRA-2 data varies obviously with latitude and decreases from the equator to both poles, especially in the Southern Hemisphere. Moreover, the RMSs of most of the IGS stations in the middle- and high-latitude regions are less than 1.5 cm, while the RMSs in the low-latitude regions are relatively large, but the RMSs are still less than 2.5 cm. The reasons are given above.

### 3.2. Accuracy Comparison of the MERRA-2 ZWD/ZTD and Radiosonde Data

To further verify the accuracy of the ZWD and ZTD values calculated from the MERRA-2 data, the 12 h time resolution profile data of 409 radiosonde stations around the world in 2015 were used to test the accuracy of the ZWD and ZTD values calculated from MERRA-2 data. First, the ZWD and ZTD data of each radiosonde station in the world at 0:00 and 12:00 UTC are calculated, and then the daily average bias and RMS of the ZWD and ZTD values at each radiosonde station calculated by the integration of the MERRA-2 data are obtained. Finally, descriptive statistics of the annual average bias and RMS of the ZWD and ZTD values calculated from the MERRA-2 data of each radiosonde station are obtained. The results are shown in Table 3 and Figure 9.

Table 3 shows that the global biases of the ZWD and ZTD values calculated from the MERRA-2 data range from −2.41 cm to 3.64 cm and −2.66 to 4.59 cm, respectively, and the average bias values are 0.47 cm and 0.46 cm, respectively. It can be seen from these results that there is a larger absolute bias appears in the calculation of the ZTD, but its average global bias is still small. In terms of the RMS, the ranges of the variations are 0.04–4.50 cm and 0.37–4.70 cm, with averages of 1.36 cm and 1.44 cm, respectively. Although the maximum RMS is 4.70 cm when calculating the ZTD, the MERRA-2 data yield a smaller average RMS in the global calculation of the ZWD and ZTD, which indicates that the MERRA-2 data have a high accuracy in the global calculation of the ZWD and ZTD. As seen from Figure 9, the biases of the ZWD values calculated from the MERRA-2 data are smaller mainly in the middle- and high-latitude regions, and the radiosonde stations in some of the low-latitude regions, such as South Asia, eastern Pacific, north-central Africa, and northern South America, have relatively large positive bias values, indicating that the ZWD values calculated from the MERRA-2 data in these areas are too large. The ZTD still shows smaller bias values in the middle- and high-latitude regions worldwide, and relatively large negative biases appear at some of the radiosonde stations in southern Europe and central South America, which indicates that the ZTD values calculated from the MERRA-2 data are relatively small in these regions, while relatively large positive biases exist in some radiosonde stations located in low-latitude South Asia and the eastern Pacific Ocean; this finding indicates that the ZTD values calculated from the MERRA-2 data are relatively large in these regions. The RMSs of the ZWD and ZTD values estimated by the MERRA-2 data are relatively small in the middle- and high-latitude regions of the world, while the RMSs of the ZWD in low-latitude regions such as South Asia, the Eastern Pacific and north-central Africa are relatively large. The ZTD values have relatively large RMSs in low-latitude south Asia, the Eastern Pacific, northwest Africa and southern Europe. The reason may be consistent with the results of IGS station verification analysis, that is, the low-latitude region is affected by complex climate systems such as tropical rainforest, tropical ocean, and savanna climates. It is further shown that the global calculation of the ZWD and ZTD information by the MERRA-2 data has high accuracy and high stability.

To analyze the daily average variations in the bias and RMS of the ZWD and ZTD values calculated from the MERRA-2 data, six representative radiosonde stations, namely, stations 04018, 89512, 54857, 94866, 91334 and 88224, were selected in the high-, middle- and low-latitude regions of the Northern and Southern Hemispheres. The daily average bias and RMS are calculated, and the results are shown in Figure 10, Figure 11, Figure 12 and Figure 13 and Table 4.

From Figure 10 and Figure 11, Table 4, it can be seen that the average biases of the ZWD and ZTD values calculated from the MERRA-2 data in middle- and high-latitude regions are smaller throughout the year. Although the daily average bias at station 89512 in the Southern Hemisphere and station 54857 in the Northern Hemisphere show certain seasonal variations, the absolute values of the daily average bias are mostly within 2.0 cm, especially in high-latitude regions, and most of them are within 1.0 cm. However, the ZWD and ZTD values calculated from the MERRA-2 data show a relatively large daily average biases in low-latitude regions without obvious seasonal variations, mainly due to the influence of the complex climate in these regions. Nevertheless, the absolute bias values are less than 4.0 cm for most of the year. Figure 12 and Figure 13, Table 4 show that the ZWD and ZTD values calculated from the MERRA-2 data at middle and high-latitude regions in the Northern Hemisphere show smaller average RMSs throughout the year, but station 54857 in a middle-latitude region has relatively large RMSs in summer. The RMSs in the middle- and high-latitude regions of the Southern Hemisphere are relatively small throughout the year, showing relatively large RMSs in winter and remaining within 2.0 cm overall. The main reason is that most of the middle- and high-latitude regions in the Southern Hemisphere are oceans, and the variations in the meteorological parameters in this area are relatively stable. The ZWD and ZTD values calculated from the MERRA-2 data in low-latitude regions show relatively large RMSs with no obvious seasonal variations. The RMS values in the Northern Hemisphere are larger than those in the Southern Hemisphere, but most of the RMS values are within 4.0 cm throughout the year for the reasons described above.

To analyze the monthly variations in the bias and RMS of the ZWD and ZTD values calculated from the MERRA-2 data, the bias and RMS of the ZWD and ZTD values are calculated for the 409 radiosonde stations worldwide according to the monthly average, and the results are shown in Figure 14.

From Figure 14, it can be seen that the monthly average bias of the ZWD and ZTD values calculated from the MERRA-2 data shows obvious seasonal variations, which shows positive bias in all months of the year. The largest bias values of ZWD and ZTD are 0.78 and 0.76 cm in the summer months, and the smallest bias of that are 0.25 and 0.24 cm in the winter months, respectively. In addition, the seasonal characteristics of the monthly average RMS calculated from the MERRA-2 data for the ZWD and ZTD values globally are similar to those of the monthly average bias. The largest RMS values of ZWD and ZTD are 1.67 and 1.77 cm occur in the summer months, and the smallest RMS values of that are 1.04 and 1.11 cm occur in the winter months, respectively, the reasons for which are described above. In conclusion, this figure further demonstrates that the MERRA-2 data have good seasonal performance in calculating the ZWD and ZTD information globally.

To analyze the variation in the bias and RMS of the ZWD and ZTD values calculated from the global MERRA-2 data, the bias and RMS of the ZWD and ZTD values calculated for the 409 radiosonde stations were statistically analyzed in terms of the elevation. The results are shown in Figure 15.

It can be seen from Figure 15 that the bias and RMS of the ZWD and ZTD values calculated from the MERRA-2 data have no obvious relationship with elevation compared with the ZWD and ZTD information calculated by the radiosonde stations. The bias of ZWD and ZTD range from −2.41 to 3.64 cm and −2.66 to 4.59 cm with mean values of 0.47 and 0.46 cm, respectively. The RMS of ZWD and ZTD range from 0.04 to 4.50 cm and 0.37 to 4.70 cm with mean values of 1.36 and 1.44 cm, respectively. However, this figure shows a relatively large bias and RMS at the radiosonde stations at elevations below 100 m. The main reason may be that the meteorological parameters are very active in low-altitude regions, and interpolation errors are easily introduced in the vertical interpolation of meteorological parameters. However, its absolute bias is within 2.0 cm, and its RMS is within 3.0 cm for most elevations. Therefore, this paper further illustrates that the ZWD and ZTD values of the four MERRA-2 grid points around the radiosonde stations can be calculated by direct integration based on the elevation of the radiosonde station as the starting height, achieving good results and significantly weakening the interpolation errors introduced by the ZWD and ZTD values in terms of the elevation.

To further analyze the variation in the bias and RMS of the ZWD and ZTD values calculated from MERRA-2 data in terms of the latitude, the bias and RMS of the ZWD and ZTD values calculated for the 409 radiosonde stations worldwide are shown in terms of the latitude, and the results are shown in Figure 16.

Figure 16 shows that the biases of the ZWD and ZTD values calculated from the MERRA-2 data are relatively stable in the middle- and high-latitude regions of the world, and the ZWD shows positive bias values for most stations, while the biases of the ZWD and ZTD values in the low-latitude regions fluctuate greatly, but the biases of most of the stations around the world vary from −2.0 to 2.0 cm. The bias of ZWD and ZTD range from −2.41 to 3.64 cm and −2.66 to 4.59 cm with mean values of 0.47 and 0.46 cm, respectively. The RMS of MERRA-2 data in the global calculation of ZWD is obviously related to the latitude and gradually decreases from the equator to the poles. The RMS of the ZTD values does not have as significant of a relationship with the latitude as that of the ZWD values. In general, the RMS of the ZTD values tends to decrease gradually from the equator to both poles. The RMS of ZWD and ZTD range from 0.04 to 4.50 cm and 0.37 to 4.70 cm with mean values of 1.36 and 1.44 cm, respectively. Moreover, the RMSs of the ZWD and ZTD values are less than 2.0 cm for most of the stations in the middle- and high-latitude regions of the world, while relatively large RMSs exist for the stations in the low-latitude regions, but the RMSs of most of the stations are still less than 3.0 cm; the reason is described above.

## 4. Conclusions

In recent years, atmospheric correction or global tropospheric delay modeling using atmospheric reanalysis data for spatial positioning has gained wide attention. In view of the fact that there is no report evaluating the accuracy of the ZWD and ZTD values calculated from the MERRA-2 data in the world, in this paper, the accuracy of the integration calculation of the ZWD and ZTD values from MERRA-2 data is evaluated by combining the precision products of IGS and radiosonde station data. This paper also proposes the direct use the altitude of the IGS and radiosonde stations as the integral starting altitude to calculate the ZWD and ZTD values of the four MERRA-2 data grid points around the IGS and radiosonde stations. The following findings are obtained:

(1) Taking the precision ZTD products of 316 IGS stations from around the world from 2015 to 2017 as the reference values, the average RMS of the ZTD values calculated from the MERRA-2 data is better than 1.35 cm, and the accuracy difference between different years is small. In terms of the time dimension, the bias and RMS of the ZTD values calculated from the MERRA-2 data show certain seasonal variations. The largest bias and RMS exist in the summer months, while the minimum bias and RMS appear in the winter months. The reason may be related to the active meteorological parameters in summer. In the spatial dimension, the bias and RMS of the ZTD values calculated from the MERRA-2 data vary significantly with latitude, especially the RMS, which gradual decreases from the equator to the poles. However, the bias and RMS of the ZTD values calculated from the MERRA-2 data do not obviously vary with elevation. The main reason is that the ZTD values of the four MERRA-2 grid points around the IGS stations are calculated directly from the elevation integrals of the IGS stations, which eliminates the error caused by ZTD interpolation in terms of the elevation.

(2) Relative to the radiosonde data, the RMS of the ZWD and ZTD values calculated from the MERRA-2 data are better than 1.37 cm and 1.45 cm, respectively. Moreover, the bias and RMS of the global calculation of the ZWD and ZTD values using the MERRA-2 data also show some temporal and spatial characteristics, which are similar to the test results of the IGS stations. It is suggested that MERRA-2 data can be used for global tropospheric vertical profile model construction because of their high accuracy and good stability in the global calculation of the ZWD and ZTD.

## Figures and Tables

**Figure 1 sensors-20-06440-f001:**
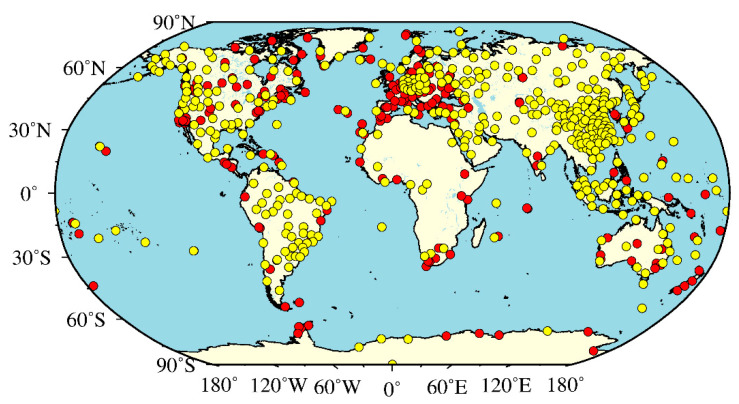
Distributions of the global International Global Navigation Satellite System (GNSS) Service (IGS) stations and radiosonde stations (red circles represent the IGS stations, and yellow circles represent the radiosonde stations).

**Figure 2 sensors-20-06440-f002:**
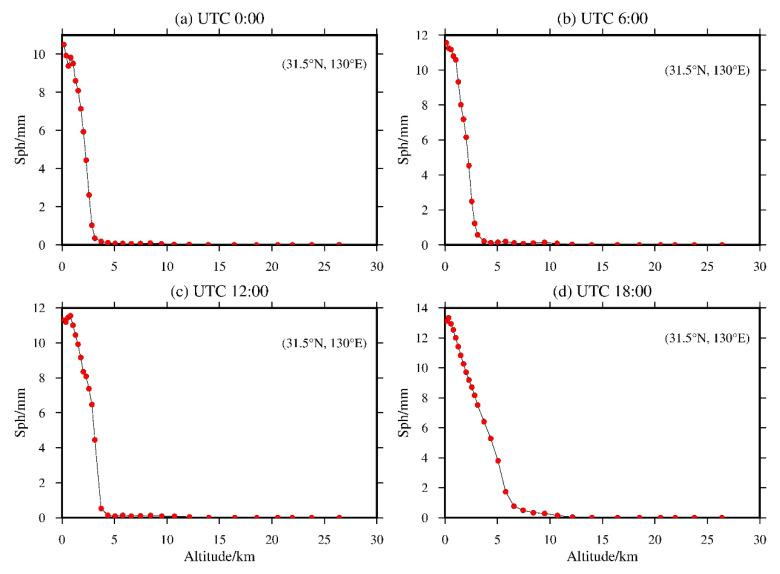
Variation in the specific humidity of the grid point (31.5° N, 130.0° E) at UTC (**a**) 0:00, (**b**) 6:00, (**c**) 12:00 and (**d**) 18:00 in the elevation direction.

**Figure 3 sensors-20-06440-f003:**
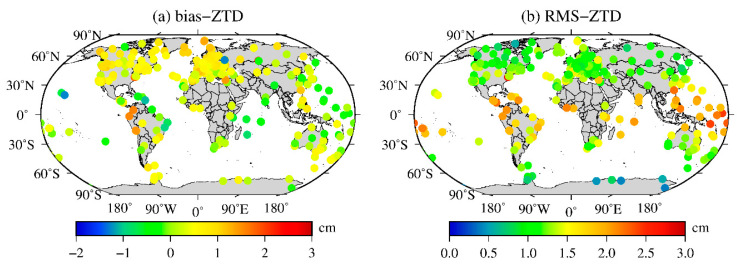
Global distributions of the annual means of the (**a**) bias and (**b**) RMS of the MERRA-2 ZTD values compared to the IGS ZTD values for 2016.

**Figure 4 sensors-20-06440-f004:**
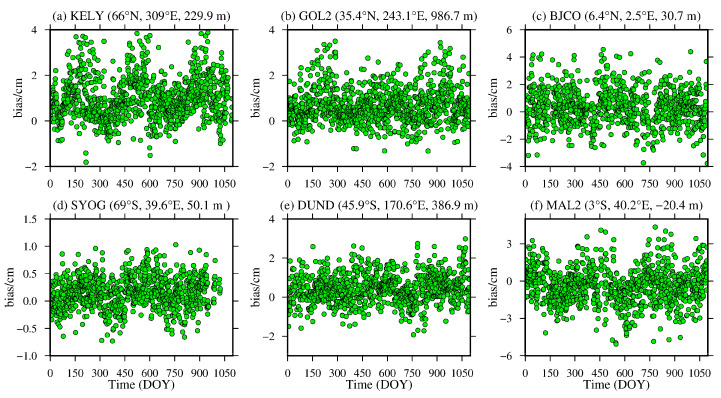
ZTD bias time series of the (**a**) KELY, (**b**) GOL2, (**c**) BJCO, (**d**) SYOG, (**e**) DUND and (**f**) MAL2 stations derived from the MERRA-2 data from 2015 to 2017 (the content in parentheses indicates the latitude, longitude and altitude of the IGS station).

**Figure 5 sensors-20-06440-f005:**
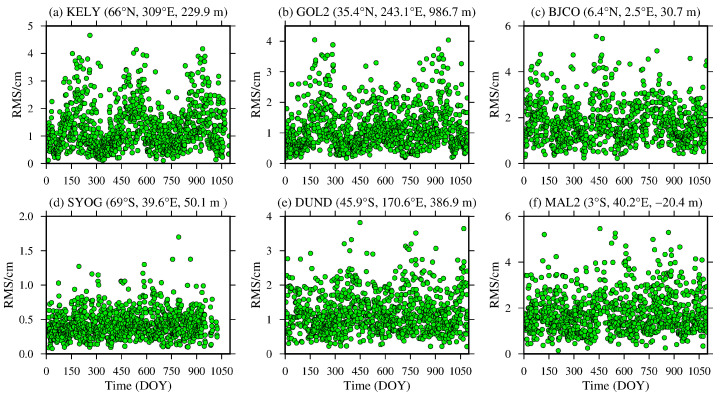
ZTD RMS time series of the (**a**) KELY, (**b**) GOL2, (**c**) BJCO, (**d**) SYOG, (**e**) DUND and (**f**) MAL2 stations derived from the MERRA-2 data from 2015 to 2017 (the content in parentheses indicates the latitude, longitude and altitude of the IGS station).

**Figure 6 sensors-20-06440-f006:**
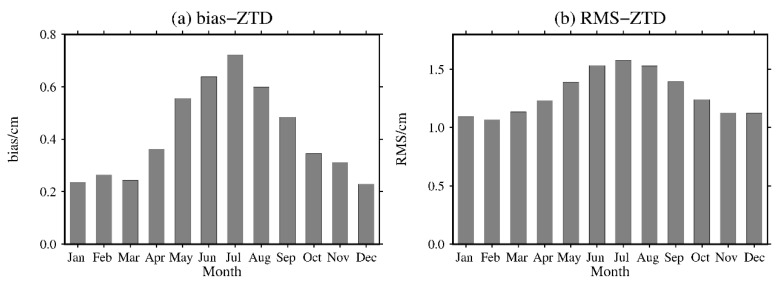
Monthly average (**a**) bias and (**b**) RMS of the MERRA-2 ZTDs at 316 GNSS stations worldwide in 2016.

**Figure 7 sensors-20-06440-f007:**
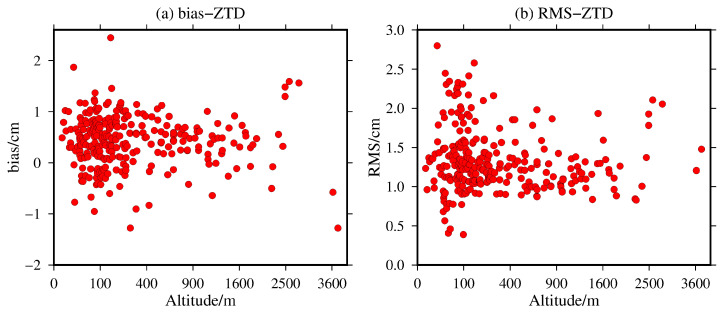
The (**a**) bias and (**b**) RMS of the MERRA-2 ZTD values with respect to altitude variations.

**Figure 8 sensors-20-06440-f008:**
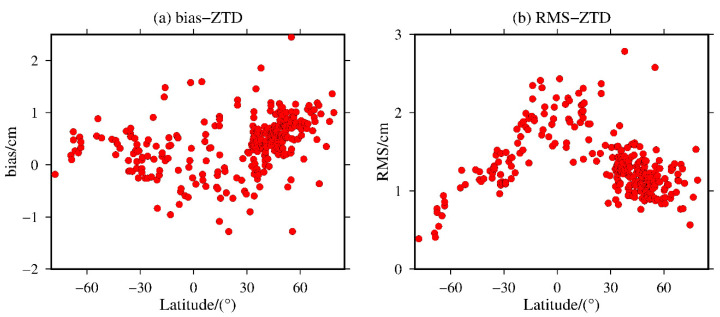
The (**a**) bias and (**b**) RMS of the MERRA-2 ZTD values with respect to latitude variations.

**Figure 9 sensors-20-06440-f009:**
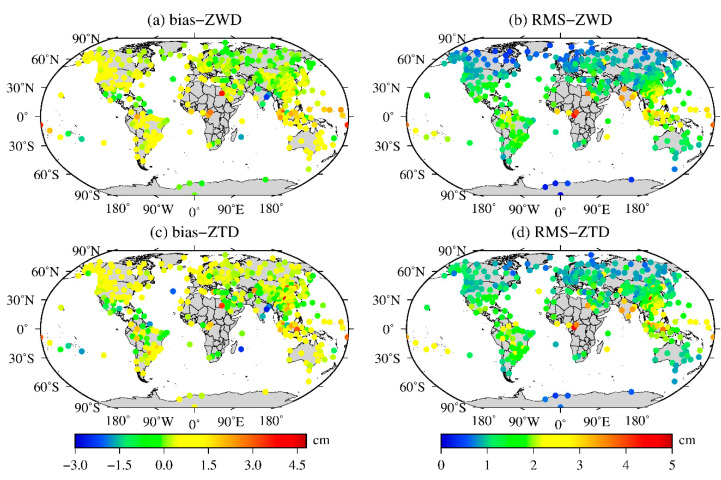
Global distributions of the annual average (**a**,**c**) bias and (**b**,**d**) RMS of the ZWD and ZTD values from the MERRA-2 data compared to the radiosonde data from 2015.

**Figure 10 sensors-20-06440-f010:**
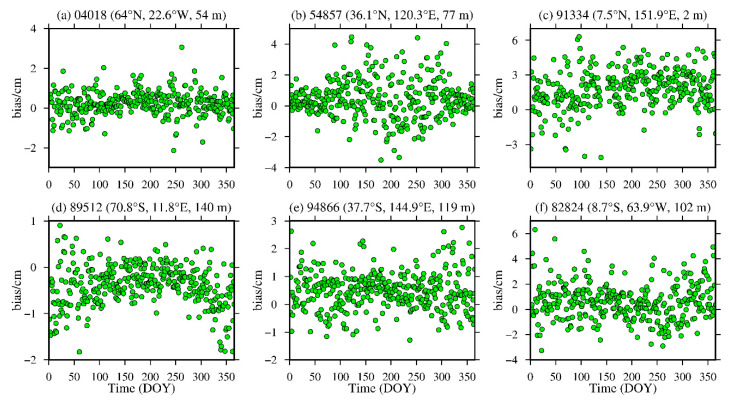
ZWD bias time series of stations (**a**) 04018, (**b**) 54857, (**c**) 91334, (**d**) 89512, (**e**) 94866 and (**f**) 82824 derived from the MERRA-2 data from 2015 (the content in parentheses indicates the latitude, longitude and altitude of the radiosonde station).

**Figure 11 sensors-20-06440-f011:**
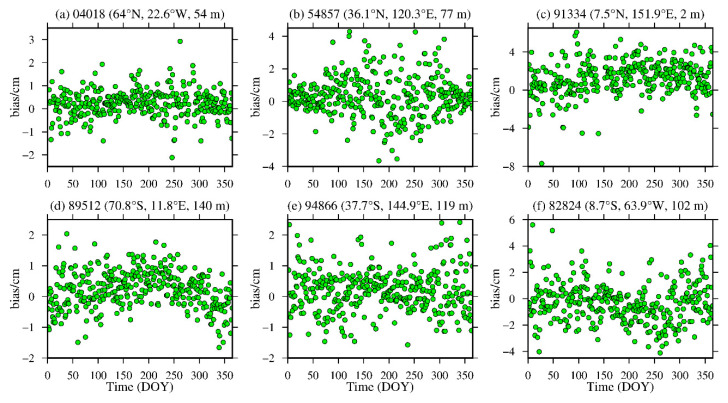
ZTD bias time series of stations (**a**) 04018, (**b**) 54857, (**c**) 91334, (**d**) 89512, (**e**) 94866 and (**f**) 82824 derived from the MERRA-2 data from 2015 (the content in parentheses indicates the latitude, longitude and altitude of the radiosonde station).

**Figure 12 sensors-20-06440-f012:**
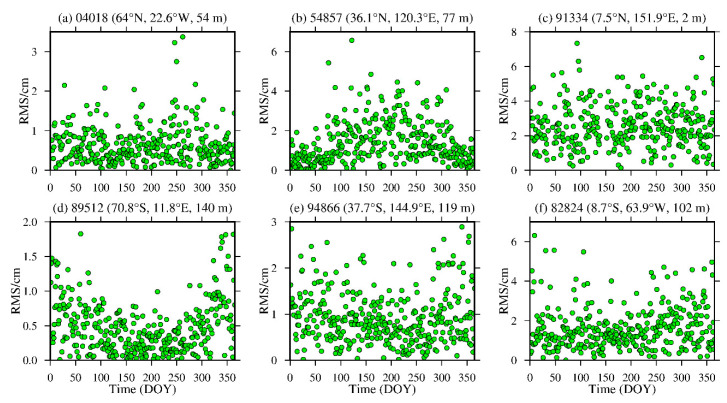
ZWD RMS time series of stations (**a**) 04018, (**b**) 54857, (**c**) 91334, (**d**) 89512, (**e**) 94866 and (**f**) 82824 derived from the MERRA-2 data from 2015 (the content in parentheses indicates the latitude, longitude and altitude of the radiosonde station).

**Figure 13 sensors-20-06440-f013:**
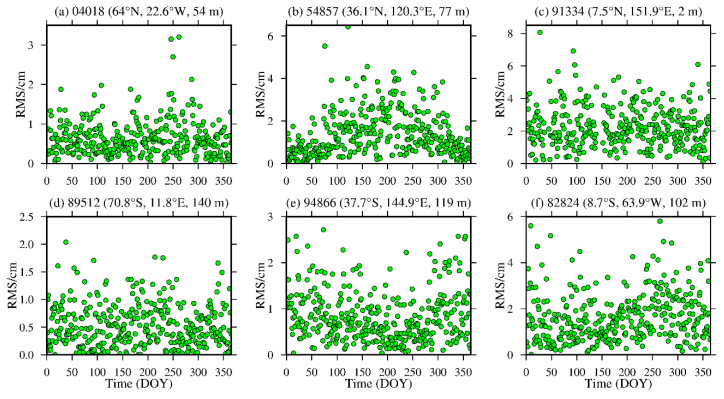
ZTD RMS time series of stations (**a**) 04018, (**b**) 54857, (**c**) 91334, (**d**) 89512, (**e**) 94866 and (**f**) 82824 derived from the MERRA-2 data from 2015 (the content in parentheses indicates the latitude, longitude and altitude of the radiosonde station).

**Figure 14 sensors-20-06440-f014:**
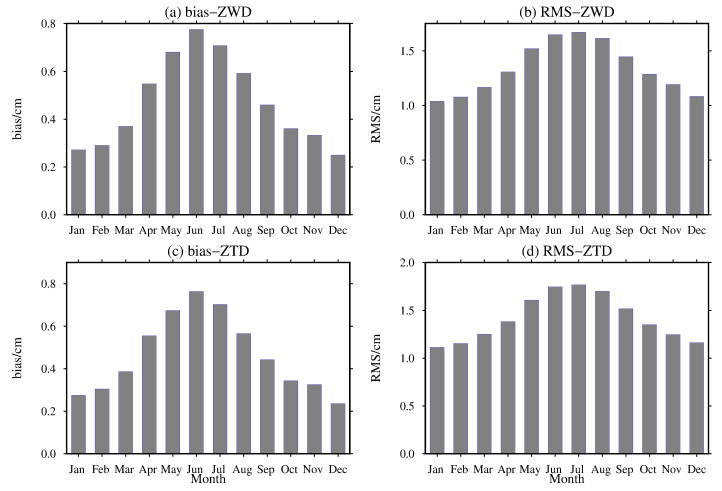
Monthly average distributions of the (**a**,**c**) bias and (**b**,**d**) RMS of the ZWD and ZTD values from the MERRA-2 data at the 409 radiosonde stations worldwide in 2015.

**Figure 15 sensors-20-06440-f015:**
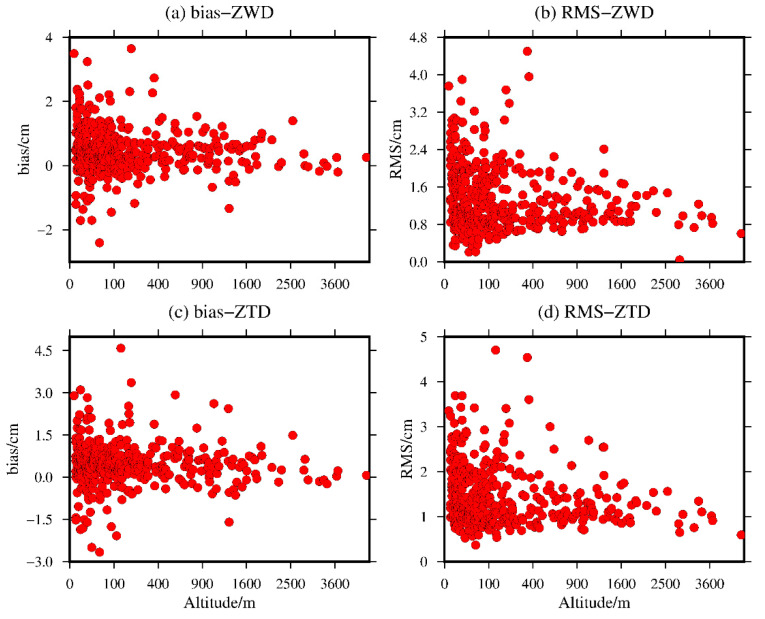
The (**a**,**c**) bias and (**b**,**d**) RMS values of the ZWD and ZTD values from the MERRA-2 data with respect to altitude variations.

**Figure 16 sensors-20-06440-f016:**
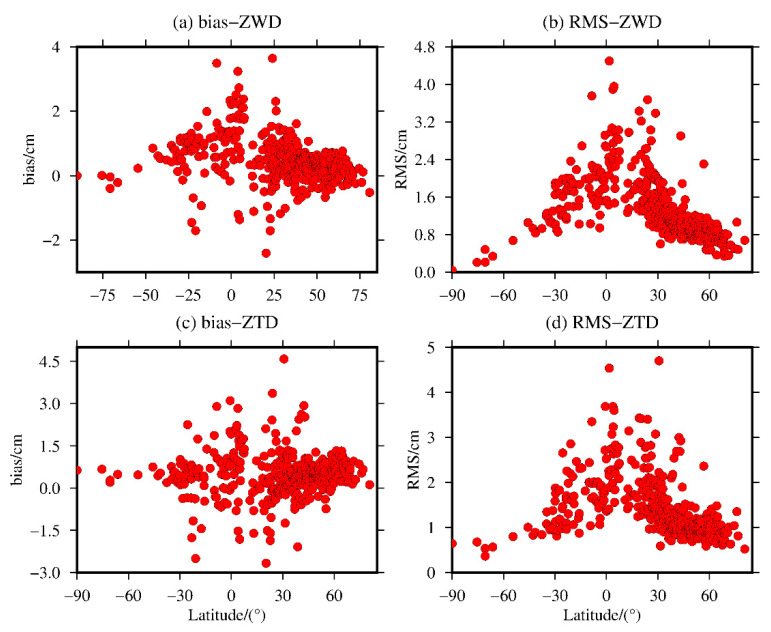
The (**a**,**c**) bias and (**b**,**d**) RMS values of the ZWD and ZTD values from the MERRA-2 data with respect to latitude variations.

**Table 1 sensors-20-06440-t001:** Descriptive statistics of the bias and root mean square (RMS) for the Modern-Era Retrospective analysis for Research and Applications (MERRA-2) zenith tropospheric delay (ZTD) tested using the IGS ZTD for 2015 to 2017.

Year	Bias/cm			RMS/cm		
	Min	Max	Mean	Min	Max	Mean
2015	−1.49	1.96	0.41	0.44	2.6	1.28
2016	−1.28	2.47	0.43	0.39	2.9	1.35
2017	−1.28	2.28	0.5	0.42	2.67	1.34

**Table 2 sensors-20-06440-t002:** Descriptive statistics of the daily average bias and RMS of six representative IGS stations in 2016.

Station Name	Bias/cm	RMS/cm
KELY	0.99 [−1.82, 3.88]	1.31 [0.11, 4.66]
GOL2	0.69 [−1.32, 3.50]	1.17 [0.18, 4.04]
BJCO	0.35 [−3.78, 4.56]	1.80 [0.23, 5.55]
SYOG	0.15 [−0.73, 1.03]	0.44 [0.08, 1.70]
DUND	0.44 [−1.93, 2.98]	1.21 [0.21, 3.82]
MAL2	−0.38 [−5.07, 4.35]	1.80 [0.14, 5.46]

**Table 3 sensors-20-06440-t003:** Descriptive statistics of the bias and RMS of the zenith wet delay (ZWD) and ZTD values derived from the MERRA-2 data and compared to the radiosonde data from 2015.

	Bias/cm			RMS/cm		
	Min	Max	Mean	Min	Max	Mean
ZWD	−2.41	3.64	0.47	0.04	4.5	1.36
ZTD	−2.66	4.59	0.46	0.37	4.7	1.44

**Table 4 sensors-20-06440-t004:** Descriptive statistics of the daily average bias and RMS of the ZWD and ZTD values of six representative radiosonde stations from 2015.

	ZWD		ZTD	
Station Name	Bias/cm	RMS/cm	Bias/cm	RMS/cm
4018	0.24 [−2.14, 3.06]	0.65 [0, 3.37]	0.24 [−2.12, 2.92]	0.65 [0.01, 3.20]
54857	0.44 [−3.51, 4.46]	1.40 [0.01, 6.57]	0.32 [−3.65, 4.28]	1.38 [0.03, 6.44]
91334	1.80 [−4.13, 6.30]	2.57 [0.13, 7.34]	1.27 [−7.69, 6.07]	2.28 [0.14, 8.06]
89512	−0.40 [−1.83, 0.90]	0.48 [0, 1.83]	0.21 [−1.66, 2.04]	0.54 [0, 2.04]
94866	0.48 [−1.29, 2.76]	0.93 [0.02, 2.89]	0.20 [−1.57, 2.41]	0.84 [0.04, 2.71]
82824	0.57 [−3.27, 6.31]	1.64 [0, 6.31]	−0.32 [−4.11, 5.60]	1.65 [0.01, 5.80]

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
