# Peer review of "Evaluation of the ZWD/ZTD Values Derived from MERRA-2 Global Reanalysis Products Using GNSS Observations and Radiosonde Data"

_sensors, 2020, doi:10.3390/s20226440_

Round 1

Reviewer 1 Report

  1. In section 2.1, the author mentioned that two sets of ZTD can be obtained by radiosonde data and GNSS, and both of them can be used as reference values. It is better to compare the two groups of reference values, analyze their advantages and disadvantages, and improve the persuasiveness of the reference values.
  2. Why can the influence of elevation be eliminated or weakened by calculating the ZWD and ZTD information of the nearest four grid points of each station with the height of GNSS and radiosonde station as the starting height of integration?
  3. From Figures 4 and 5, how to distinguish seasons by horizontal axis? The author may be able to mark four seasons or days on the horizontal axis as shown in Figure 10.
  4. How to calculate the ZTD as a reference value from the pressure and ZWD derived from the radiosonde?
  5. ZWD experimental results lack of reference value, readers cannot get the accuracy improvement range of ZWD. In section 3.1, the accuracy of the global GGOS atmospheric grid ZTD products is used as the reference value in ZTD experiment, the author can select the accuracy of a widely used ZWD calculation method as a reference value to obtain the accuracy improvement rate.

Reviewer 2 Report

Tropospheric delay is an important error source in GNSS, and the water vapor retrieved from tropospheric delay is widely used in meteorological research such as climate analysis and weather forecasting. This manuscript evaluates the MERRA-2-derived ZWD/ZTD using GNSS and radiosonde data. The description of data and methodology is clear, and its results and associated analysis are demonstrative and convincing. The derived findings are well supported by the relatively rich data-based study, and the presentation of the manuscript is well delivered through the text and illustrations.

The English of this manuscript can be improved, and some minor comments are listed below.

  1. The first paragraph in the Introduction part gave lots of conclusive statement but only had two references. Please add references to support your statement.
  2. Line 74-76, please define “meaningful conclusion”.
  3. Line 83-87, please add reference(s).
  4. Line 130-131, what did the authors mean by say “comparable results”.
  5. Line 153-157, can you show some supportive results or explain more?
  6. Section 3.2. The accuracy comparison of MERR-2 ztd/zwd with radiosonde is very similar to that with GNSS, I suggest the authors should shorten the similar results while focus on analyzing the difference between radiosonde and GNSS.

Reviewer 3 Report

Dear Editor,

my comments for Authors:

  1. all acronyms must be explained in Abstract, e.g. RMS
  2. all acronyms must be explained in main body of text, please check it.
  3. what is a source of Equations (1-5)? Please add few position in Reference.
  4. Equation (7), what means "H"? It is a ellipsoidal height or orthometric height?
  5. Figure 2. Please better describe the results from Figure 2.
  6. what is a source of Equations (8-9)? Please add few position in Reference.
  7. Figures 4 and 5. Please better describe the results from Figures 4 and 5, e.g. min, max and mean values.
  8. Figure 6. Please better describe the results from Figure 6, e.g. min, max values.
  9. Figures 7 and 8. Please better describe the results from Figures 7 and 8, e.g. min, max and mean values.
  10. Figures 10-13. Please better describe the results from Figures 10-13, e.g. min, max and mean values.
  11. Figure 14. Please better describe the results from Figure 14, e.g. min, max values.
  12. Figures 15-16. Please better describe the results from Figures 15-16, e.g. min, max and mean values
  13. References, see comment 3 and 6.

The paper is good, after minor correction I accept the paper.
